# Can Health-Enhancing Sporting Programs in Sports Clubs Lead to a Settings-Based Approach? An Exploratory Qualitative Study

**DOI:** 10.3390/ijerph18116082

**Published:** 2021-06-04

**Authors:** Linda Ooms, Mette van Kruijsbergen, Dorine Collard

**Affiliations:** Mulier Institute, P.O. Box 85445, 3508 AK Utrecht, The Netherlands; mettevk.91@gmail.com (M.v.K.); d.collard@mulierinstituut.nl (D.C.)

**Keywords:** sports club, sporting program, health-enhancing physical activity, settings-based approach, inactive, stimulating factors, hindering factors, qualitative study

## Abstract

There is limited knowledge about how a settings-based approach can be best applied in a sports club setting. This qualitative exploratory study examined whether and how sporting programs focusing on individual behavior change (i.e., increasing physical activity levels of inactive people) and implemented on the micro-level of the sports club, can be a first step towards a settings-based approach (i.e., inclusion of the meso- and macro-level of the sports club). In addition, this study explored factors that influenced the inclusion of the meso- and macro-level of the sports club. Telephone interviews were conducted with representatives of sixteen sports clubs about program activities on all levels of the sports club. Thematic analyses were performed to explore stimulating and hindering factors. After multiple years, six sports clubs also had program activities on the meso-level and twelve sports clubs had activities on the macro-level. Program activities differed per level within a sports club and on the same level between sports clubs. Cultural and social factors influenced macro-level activities, while predominantly economic factors influenced meso-level activities. Based on these factors, sports clubs could develop, prioritize, and choose strategies that support them in developing a settings-based approach when increasing physical activity levels of inactive citizens.

## 1. Introduction

Regular participation in physical activity is important for the physical, mental, and social well-being of people of all ages [1,2,3,4]. Nonetheless, almost a third of the world population (18 years and older) and half of the Dutch population (4 years or older) is insufficiently active to realize these benefits [5,6]. These people are at an increased risk of different health problems and mortality, especially the ones that are completely inactive [3,4]. To promote public health, effective strategies promoting physical activity specifically focusing on inactive people are therefore needed.

To stimulate and maintain a healthy and active lifestyle, strategies should not only focus on changing individual behaviors, but also on changing the environment, so that the environment supports the desired behavior [7]. In this regard, the World Health Organization (WHO) points out the importance of a settings-based approach in promoting healthy behaviors, including a physically active lifestyle [8,9]. A setting is “the place or social context in which people engage in daily activities in which environmental, organizational and personal factors interact to affect health and well-being” [10]. The settings-based approach is based on social-ecological theory and emphasizes the importance of the whole system of the setting (i.e., influencing factors on the intrapersonal, interpersonal, institutional, community, and policy level) rather than just individual responsibility (i.e., intrapersonal factors) when stimulating healthy behaviors. Important key elements of the approach are: (1) creating supportive and healthy environments to make “the healthy choice the easy choice”; (2) integrating health promotion in the daily activities of the setting; and (3) developing links with other settings and the wider community [7,11,12,13].

The settings-based approach to health promotion has been applied to multiple settings, such as schools, workplaces, and prisons [11,12,13]. More recently, policy makers, researchers, and professionals working in the health and sport sectors acknowledged the potential of sports clubs in promoting healthy behaviors, including a physically active lifestyle among inactive people [14,15,16,17,18,19,20,21]. In this article, a sports club is defined as a non-profit, voluntary organization that offers opportunities for active sport participation (i.e., recreational and/or competitive sports, individual and/or team sports) to its members. The key characteristics are voluntary memberships and organizational goals that are oriented to members’ interests, democratic and participatory decision-making processes, voluntary engagement, and autonomy in its operation [22]. Due to their wide reach into the community, social and informal learning environment, sports clubs have great potential to promote a healthy and active lifestyle in the population [19,23]. Although sports clubs have the potential to stimulate physical activity among inactive citizens, they also face challenges. Inactive people may experience different personal and social barriers, which could prevent them from participating in sport and physical activities [24,25,26]. Moreover, the high prevalence of sport injuries and drop-out among novice sport participants is of concern [27]. Furthermore, sports clubs’ focus is on providing training and competitions. Promoting physical activity among inactive target groups can be seen as an extra burden and lack of time, human resources (i.e., the dependence on volunteers), and skills (e.g., to attract and guide inactive people) can hinder physical activity promotion among this target group at sports clubs [14,15,16,17,21].

The theory of the settings-based approach to health promotion has been specifically adapted to the sports club setting [19,28]. Within this theory, there are four types of determinants: (1) cultural determinants: the values associated with the position of health promotion within a sports club setting and in the club’s policies and operational principles; (2) social determinants: the social attitude of sports club’s actors with regard to health promotion; (3) economic determinants: the financial and practical (time and knowhow) resources to realize health promotion; and (4) environmental determinants: the environmental conditions in which daily activities take place. These determinants influence activities on three levels of the sports club setting: the macro- (overall policies and orientations of a club), meso- (guidance activities of club leaders), and micro-level (trainers’ activities in guiding, altering, or supporting actions of sport participants). When promoting physical activity among inactive target groups from a settings-based perspective, it means that activities are performed on all levels of the sports club, where activities on the higher levels (meso- and macro-level) support activities to increase physical activity of inactive people on the lower levels (micro- and meso-level). However, a recent systematic mapping review found that many health promotion interventions in sports clubs, including those aimed at increasing physical activity, did not yet apply a settings-based approach. Most interventions focused on activities on one level of the sports club (i.e., the micro- or macro-level) and on changing individual behaviors rather than the club overall [29]. 

The theoretical development concerning the application of the settings-based approach in the sports club setting is relatively recent and there is still limited knowledge about how the settings-based approach can be best applied in this setting. The theory itself provides limited guidance on where to start (i.e., on which sports club level(s) and how) [19,28,30]. Directly starting with program activities on all sports club levels may not be feasible since sports clubs are settings that are principally based on volunteer work with a core business centered on sports [29]. There are guidelines for health promotion activities in youth sports clubs. These suggest starting with macro-level activities, such as determining health promotion aims and turning these into a written form (i.e., policy development) [29,31]. In the aforementioned mapping review, it is also suggested that interventions focusing on individuals or individual behavior can be a first step to achieve a settings-based approach and that intervention success, including benefits for the club (e.g., more club members, better image of the club), can convince others, such as trainers, board members, local organizations, and potential participants, to participate [29]. Consequently, starting with sporting programs for inactive people on the micro-level, which is more closely related to sports club’s core business (i.e., providing sport activities vs. policy development) might be a feasible first step to develop to a settings-based approach. Therefore, this study aims to explore whether and how sporting programs focusing on individual behavior change and implemented in sports clubs, can be a first step towards a settings-based approach in the sports club’s context. We hypothesize that the sports clubs, starting with sport activities for inactive people at the micro-level, have included multiple levels of the sports club setting to successfully sustain these programs in the long-term. Furthermore, we will explore factors that have facilitated or hindered the inclusion of the meso- and macro-level of the sports club.

## 2. Materials and Methods

A qualitative study design was used that included Dutch sports clubs that have been providing sporting programs for inactive people over a period of multiple years. Program activities and influencing factors were (retrospectively) assessed six and a half years after a funded implementation period. This study was part of a larger study in which the long-term sustainability of the programs was examined [32]. The sporting programs were developed by National Sports Federations (NSFs) within the National Action Plan for Sport and Exercise (NAPSE). The NAPSE was a national program aimed at increasing the number of Dutch inhabitants that were sufficiently active. Ten NSFs received funding for a period of three years (2008–2011) to implement a total of fourteen sporting programs (three NSFs implemented multiple programs) in their associated sports clubs. After the funding period, the sporting programs had to be self-sustaining [32]. 

### 2.1. Sample

For the research, the program coordinators of the ten NSFs were asked to provide the e-mail addresses of two sports clubs that had started the program between 2008–2011 and still provided the program at the moment of the research. The three NSFs that implemented multiple programs had to deliver contact details of sports clubs for each program. When a sports club agreed to participate, the NSF provided the email address of the coordinator of the program within the sports club (mostly a trainer) to the researchers. Subsequently, an invitational email was sent to the sports club to participate in the study. Not all NSFs provided enough contact details. Four NSFs provided no contact details at all, because they stopped (coordinating) the program at the NSF level (*n* = 3) or just did not respond (after different reminders) to our request (*n* = 1). However, one NAPSE sporting program was continued by sports clubs without involved of the NSF. Two sports clubs providing this program were retrieved through searching the internet. One contact person was excluded, because this person was not directly involved in the continued activities. This led to the inclusion of sixteen sports clubs that continued a total of ten different NAPSE sporting programs. A description of the sporting programs included in the study can be found in Table 1.

### 2.2. Procedures

A total of sixteen semi-structured telephone interviews of 45–60 min duration were conducted with representatives of sports clubs between January and March 2018. An interview guide was used based on a pre-specified sustainability framework [32]. Questions concentrated on the course of the program during the past years and (changes in) program activities on the different levels of the sports club. In addition, the interview guide contained questions about the current target group, the results of the program for the sports club and future plans for the program. Furthermore, background information about the respondent and sports club were obtained [32]. The interview guide can be found in Appendix A. The interviews were performed by two researchers (L.O., M.V.K.). Interviewing was practiced so that interviewing was done in the same manner. All interviews were recorded with a digital voice recorder and transcribed verbatim. 

The respondents received information about the background and aims of the study, both written (through email) and verbally (i.e., before the start of the telephone interview). Furthermore, it was explained that their participation was voluntary, all information collected during the interview would be kept strictly confidential and only anonymized data would be published. All respondents gave informed consent for their participation as well as consent for their interviews to be recorded and transcribed verbatim. Participants were not subjected to procedures, nor were they forced to follow certain behavioral rules. Thus, in agreement with the Dutch Medical Research Involving Human Subjects Act (WMO), medical ethics committee’s approval was not needed for performing this research [33]. Privacy procedures of this study were in agreement with General Data Protection Regulations [34]. The Relevancy, Appropriateness, Transparency, and Soundness (RATS) guidelines guided reporting of results [35].

### 2.3. Data Analysis

Data were analyzed using MAXQDA version 10.0 software by one researcher (L.O.). All interview transcripts were first read to become familiar with the data. Then, per transcript, program activities were classified according to the different levels of the sports club setting:-Micro-level: activities performed by the trainers to guide inactive participants, such as the sport activities offered. Moreover, education opportunities for trainers that specifically focused on (guiding) inactive people were categorized on this level.-Meso-level: activities managed by the leading persons in a club, such as representatives of the board and committees and other club leaders, to guide and support trainers’ activities.-Macro-level: policy activities within the club and activities concerning the broader community environment (e.g., collaboration with partners) to support the provision of the sporting program on the other levels.

In addition, thematic analysis was performed to explore factors that have stimulated or hindered the inclusion of the meso- and macro-level of the sports club setting. Therefore, the following steps were taken as recommended by Nowell et al. (2017) [36]. First, each transcript was read again and codes (subthemes) were inductively added. After initial coding of all transcripts, the codes (subthemes) were sorted into main themes and refined. Subsequently, the coding of all transcripts was again checked by the researcher. Thereafter, the main themes or factors were categorized by the sports club level that they influenced (i.e., meso- or macro-level; see above) and per type of determinant or factor [19,28]:-Cultural factors: factors related to values associated with the position of physical activity promotion among inactive people within a sports club setting and in the clubs’ policies and operational principles.-Social factors: factors concerning the social attitude of sports club’s actors with regard to promoting physical activity among inactive people. -Economic factors: factors related to the financial and practical (time and knowhow) resources to promote physical activity among inactive people.-Environmental factors: factors related to the environmental conditions in which the daily activities for (inactive) participants take place.

When coding was complete, cross-case analyses were performed with the individual sports clubs representing the cases [37]. For this purpose, factors were compared between sports clubs to find common stimulating and hindering factors to include the meso- and macro-level of the sports club when providing the sporting program. For instance, multiple respondents indicated that a lack of financial resources hindered supportive activities of club leaders. This was summarized as the hindering economic factor “lack of financial resources”, which acted on the meso-level. The classification of activities per level and the retrieved factors were discussed with the research team (M.V.K., D.C.). This only led to minor adjustments in classification of activities and factors. 

## 3. Results

### 3.1. Characteristics of Respondents and Included Sports Clubs

Characteristics of the respondents can be found in Table 2. Most respondents were trainers within the sports club that were involved with the program for multiple years. Characteristics of the included sports clubs can be found in Table 3. The included sports clubs varied regarding size and location in the Netherlands. The average number of participants in the sporting programs varied from 6 to 1.625 per year, with 2% to 100% of participants becoming member of the sports club each year.

### 3.2. Levels of the Sports Club Involved in Providing the Sporting Program

We explored whether sporting programs focusing on individual behavior change (i.e., sport activities provided to inactive people by trainers on the micro-level) could lead to (supporting) activities on the meso- and macro-level of the sports club. Table 4 presents the levels of the sports club that implemented program activities six and a half years after the funding period ended. Next to micro-level activities, six sports clubs included program activities on the meso-level and twelve sports clubs had activities on the macro-level. Six sports clubs had activities on both the meso- and macro-level of the sports club. For four sports clubs, program activities remained at the micro-level.

### 3.3. Activities per Level of the Sports Club

To get more insight in how each level of the sports club was involved, we also examined the type of activities performed per level per sports club. The amount and type of activities differed per level of the sports club and on the same level between sports clubs. For a detailed description of program activities per level and per sports club, see Appendix A. A short description of frequently mentioned activities on each level are described below.

#### 3.3.1. Activities on the Micro-Level

Six and a half years ago, the sporting programs started with activities on the micro-level. Activities on this level consisted of the provision of sport activities to (inactive) participants by (certified or uncertified) trainers (see also Table 1). In addition, the trainers often provided social activities to participants such as drinking coffee or tea during or after the training, but also travelling to and participating in a sport event together (i.e., with the trainer and other participants). Moreover, trainers participated in educational activities specifically to learn to guide the (inactive) target group, sometimes in the form of train-the-trainer courses in which trainers educated other (new) trainers. 

During the years, program activities were also integrated in the meso- and macro-level of the sports club: 

#### 3.3.2. Activities on the Meso-Level

After the start of the sporting programs, different club leaders, mostly one person per club, got involved in providing the program, such as a secretary or a chairman. In some cases, there was even a committee initiated specifically to guide and support trainers’ activities. Club leaders performed coordinating activities, such as coordinating promotional activities for the program, selecting trainers and volunteers and facilitating educational activities for trainers. In most cases, the coordinator had a dual role and was also (head) trainer on the micro-level.

#### 3.3.3. Activities on the Macro-Level 

Finally, different sports clubs even integrated the program into their long-term policy. In this way, financial and human resources to provide the sporting program were secured. Sports clubs also collaborated with partners in the community, such as sport stores, a senior club, the municipality, and neighborhood sport coaches to promote the program, recruit (inactive) participants, and provide the activities. Partners also provided financial (e.g., to keep participant fees low) and material (e.g., sport materials for participants) resources. NSFs facilitated sports clubs in the form of (promotional) materials, education possibilities for trainers and financial resources (to pay trainers). 

### 3.4. Factors That Influence the Inclusion of the Meso- and Macro-Level of the Sports Club

To get more insight into why some sports clubs did manage to integrate program activities on the meso- and macro-level of the sports club and other clubs did not, stimulating and hindering factors were explored. These factors are presented in Table 5. While cultural and social factors influenced macro-level activities, economic factors seemed most important at the meso-level. The factors will be explained in more detail below.

#### 3.4.1. Factors Stimulating and Hindering the Implementation of Meso-Level Activities

To involve club leaders in program activities, it was important that club leaders were enthusiastic about the program and supported the aim of the program (i.e., increasing physical activity levels of inactive people). Club leaders became enthusiastic because they liked the activities offered, experienced positive effects with participants or got inspired by enthusiastic participants and trainers:
*“When someone is enthusiastic, this stimulates others. Both participants and volunteers are enthusiastic. The people in the committee do their activities with a lot of pleasure and love for the sport. You can see that. Participants come back because of the nice atmosphere. It [the program] is fun and sociable. The program is coordinated by a group of enthusiastic people and that influences it positively. Enthusiasm forms the basis [to support these activities]. When there is no enthusiasm, it won’t work.”*—Chairman committee club 13.

Furthermore, club leaders should have enough time and financial resources to coordinate activities. This was sometimes difficult to realize due to sports clubs’ reliance on volunteers. Consequently, limited human and financial resources hindered some sports clubs to involve club leaders to support the implementation of the program on the meso-level. In addition, often only one or two club leaders were involved. When a club leader leaves, supportive activities at the meso-level may also disappear:
*“A successor will be a challenge. I’m also politically active and in three weeks, there will be elections. If it turns out as expected, I will have less time left to spend on this sport. Then, I must hand over my tasks to someone else. Now, we do this with two board members. I have someone who assists and helps me, but this person can’t do my tasks, because he also has a regular job. So that could be a threat to continue our supportive activities.”*—Secretary club 2.

#### 3.4.2. Factors Stimulating and Hindering the Implementation of Macro-Level Activities

Respondents, especially those from clubs where activities on the meso- and macro-level were developed, felt the club had a social responsibility to contribute to the health of people in their community. These clubs specifically focused on increasing physical activity levels of inactive or less advantaged groups:
*“Our club does not focus on competition. We offer recreational sport activities. We have a social responsibility to help less advantaged people to participate in sport. This is how they see us in the community.”*—Secretary club 2.

Due to the focus on inactive people, these clubs collaborated with (different) partners in the community, such as a senior’s club or a physiotherapist, to reach and recruit inactive people. For clubs, that did not specifically focus on inactive people, collaborative partners for the program were mostly absent.

In addition, different respondents stated that the program had multiple benefits for the club as a whole, such as more club members, more skilled trainers, a better image of the club and the possibility to collaborate with partners in the community. This stimulated the integration of the program into the club’s long-term policy to secure financial and human resources. On the other hand, when there were no benefits for the club as a whole, macro-level and meso-level activities were often absent. In these cases, the program was seen as a regular sport activity for a small group of people within the club, led by one or two volunteers on the micro-level:
*“I do everything myself. I’m the coordinator, but they [the participants] also call me the manager. I also provide the training sessions to the participants. That’s possible because there is only one training session per week. I have done this from the beginning.”*—(Uncertified) trainer club 7.

## 4. Discussion

Sports clubs are seen as a promising setting to stimulate physical activity among inactive people [14,15,16,17,18,19,20,21]. Regarding promoting healthy behaviors, settings-based approaches are advocated [8,9]. This qualitative study explored whether and how sporting programs focusing on individual behavior change and implemented in sports clubs, can be a first step towards a settings-based approach in the sports club’s context. Moreover, factors stimulating and hindering the inclusion of the meso- and macro-level of the sports club were explored. For this purpose, sixteen Dutch sports clubs that had been providing sporting programs for inactive people for multiple years were studied. 

The results of our study showed that the implementation of the sporting programs (micro-level activities) predominantly led to additional activities on the macro-level of the sports club. In a systematic mapping review concerning health promotion interventions in sports clubs, it was also found that intervention activities were mainly on the micro- and macro-level and to a lesser extent on the meso-level of the sports club [29]. Furthermore, our study found that mostly only one club leader was involved that had a dual role (i.e., club leader and trainer) within the club. The reliance on one club leader for the program could be a threat, because when this person leaves, support at the meso-level may also disappear. Therefore, strategies to involve multiple club leaders seem necessary.

Furthermore, our study results showed that there were differences in the amount and type of activities that were implemented on each level of the sports club. A study that examined the application of the settings-based approach in French sports clubs when implementing health promotion interventions supports these findings. The results also showed that a settings-based approach was applicable to sports clubs, but that sports clubs implemented different intervention activities per club level to realize this [38]. Consequently, there may not be a “one-size-fits-all approach” for sports club to develop a settings-based approach. 

In addition, the results of our study indicated that sporting programs aimed at individuals can lead to activities on both the meso- and macro-level of the sports club. However, to realize this, certain pre-conditions must be met. In the conceptualization of the settings-based approach in sports clubs, these pre-conditions are the cultural, social, economic, and environmental determinants that can act on each level [19,28]. In our study, we found that meso-level activities were dependent on social support from club leaders and the availability of human and financial resources. A feeling of social responsibility, including a focus on inactive people, and the acknowledgment of the multiple benefits of the program for the club, stimulated macro-level activities. In a recent study, experts in the fields of sport and health defined the determinants of a health-promoting sports club more precisely [39]. The factors found in our study resemble some of these determinants, such as “my sports club offers sports for people who are inactive or sedentary” (cultural determinant, macro-level vs. a focus on inactive people); and “officials of my sports club allocate resources to health promotion” (economic determinant, meso-level vs. availability of human resources). Consequently, addressing the cultural, social, and economic factors identified in our study might be a good starting point when implementing micro-level activities and aiming for a settings-based approach. Based on the determinants of a health promoting sports club, Van Hoye et al. (2020) developed fourteen strategies and related intervention components to support the implementation of a settings-based approach to health promotion in sports clubs [40]. Some of these strategies may support in addressing the identified factors in our study after implementation of the sporting program (i.e., micro-level activities). Strategies such as ensuring internal club communication and writing down sports club’s goals regarding the promotion of physical activity among inactive people, may, respectively, raise awareness of the multiple benefits of the program for the club and create a feeling of social responsibility to offer activities for inactive people. This in turn could enhance macro-level activities, such as the integration of the program in the long-term policy of the club and collaboration with partners in the community. Other suggested strategies, such as regularly reviewing financial and human resources and including club leaders, trainers, and participants in the decision-making process, may enhance supportive activities on each sports club level, including the meso-level. As mentioned previously, only a few sports clubs implemented activities on this level. 

### 4.1. Practical Implications

Overall, the results of this exploratory study provide some insight into how and under which preconditions the meso- and macro-level of the sports club can be involved when starting with sporting programs aimed at inactive people on the micro-level. The results could guide sports clubs in developing, prioritizing, and choosing strategies that support them in developing a settings-based approach when increasing physical activity levels of inactive people in their community. Starting with a sporting program aimed at individual behavior change and, subsequently, addressing the identified stimulating and hindering factors with the aforementioned strategies, may be a good starting point. It should be noted that including the meso- and macro-level of the sports club takes time and sports clubs might need support to execute the mentioned strategies [41]. This could be a role for the NSF, since different sports clubs were facilitated by the NSF, but particularly to implement micro-level activities. The NSF could create a resource site for associated sports clubs to access (promotional) materials and good ideas regarding these topics or create other opportunities for sports clubs to share experiences. Moreover, sports clubs themselves could contact other clubs to share good practices.

### 4.2. Strengths and Limitations

This was a practice-based study, including sports clubs that implemented sporting programs aimed at inactive people in the real-life sport setting. In addition, a diverse sample of sports clubs was included in this study. These are the strengths of this study. However, there are also limitations. First, considering the qualitative nature of the study and the Dutch context, it is not known to what extent the results are generalizable to sports clubs in other countries that are differently organized. Second, due to the small sample, it was not possible to allocate the differences in implemented activities on each level to sports clubs’ characteristics, such as size or type of sport offered. Although this was not the direct aim of this study, future research including a larger sample of sports clubs could provide more insight into the influence of sports clubs’ characteristics on adapting a settings-based approach. Third, the interviews were undertaken with the coordinators of the sporting programs (mostly trainers), but other sports club actors on other levels of the club could have provided insight into the implemented activities and influencing factors from their perspective. Moreover, in the settings-based theory, it is suggested that the different levels of the setting influence the behaviors or actions of people within the setting, but that at the same time behaviors or actions of people also influence the setting [19,28]. Therefore, future research should study these sports clubs at different moments in time to provide more insight into the development process regarding a settings-based approach and interactions between (actors of) the different levels of the sports club setting. 

## 5. Conclusions

This study contributes to the understanding of how a settings-based approach could be developed within sports clubs. The results showed that sporting programs focusing on individual behavior change can lead to program activities at the meso- and macro-level of the sports club. However, to realize this, it is important to consider the identified cultural, social, and economic factors that influence meso- and macro-level activities. Based on these factors, sports club could develop, prioritize, and choose strategies that support them in developing a settings-based approach when increasing physical activity levels of inactive people in their community. Future research should focus on studying the development from micro-level to multi-level activities longitudinally, including multiple sports club actors and multiple measurements, to get more insight into the development process regarding a settings-based approach and interactions between (actors of) the different levels of the sports club setting. 

## Figures and Tables

**Table 1 ijerph-18-06082-t001:** Description of the NAPSE sporting programs included in the study.

Name of NAPSE Sporting Program (*NSF*)	Description of Sporting Program ^1^
Yakult Start to Run (*athletics*)	7-week training program for novice runners and inactive adults aimed at running 3 km continuously. Participants train 3 times a week: once under the guidance of a professional coach, and twice individually. The individual training sessions are supported by audio coaching (running app). In the final week, participants can participate in a 3 km test run. The program is offered by athletics clubs and running stores.
Judo in School (*judo*)	1–8-week program for children. Weekly judo lessons are provided at primary schools by qualified judo trainers, with possible follow-up lessons after school at the judo club.
Through 4 Days Marches (*walking*)	6-month individual training program for debutants (adults) of the Four Day March in Nijmegen. Training schedules and information can be downloaded from the internet. Participants can take part in a regional meeting (organized by local walking trainers) where they get information about their training schedule and the Four Day March in Nijmegen. In addition, they can take part in one preparatory walking event. Next to this individual training program, local walking clubs offer their own training program for the event. This includes guided training sessions and participation in different preparatory walking events.
Flexible (*gymnastics*)	8–12-week program with weekly gymnastics classes for older adults (45+ years) focusing on a specific theme (e.g., condition, power, flexibility, and coordination) at a gymnastics club.
Fit Hockey (*hockey*)	Hockey for older adults (50+ years) played in a team with soft sticks and soft balls. Training opportunities are provided on a weekly basis at the hockey club.
Thinking and Doing (*bridge*)	A two-year project in which weekly 2.5-h bridge lessons are used to create communities of older people (60+ years). After a year, physical activities can be offered (this is an optional component).
Start2Bike (*sportive cycling*)	4-week training program for novice cyclers (speed cycling, mountain biking) and inactive adults. Participants train 3 times a week: once under guidance of a professional coach, twice individually. The program is offered by (sportive) cycling clubs and cycling stores.
Cycle & Enjoy Nature (*sportive cycling*)	Weekly recreational cycling activities for older adults (45+ years) with a focus on relaxing and enjoying nature. The program is offered in the period April–October by sportive cycling clubs.
Trio-Triathlon (*triathlon*)	Organization of Trio-Triathlon events for adult participants (the three elements of a triathlon are performed by three different individuals).
Cool Moves Volley (*volleyball*)	A volleyball approach adapted to children’s needs and abilities. It is the official volleyball form to teach children 6 to 12 years the fundamentals of volleyball at volleyball clubs and the official competition form for this age group. Training opportunities are provided on a weekly basis at volleyball clubs. Clinics are provided in schools.

*NAPSE* = National Action Plan for Sport and Exercise; *NSF* = National Sports Federation. ^1^ Most recent description of the sporting programs, i.e., during the sustainability period (in 2018).

**Table 2 ijerph-18-06082-t002:** Descriptive characteristics of respondents.

Characteristic	Respondents (*n* = 16)
Gender (*n*, %)	
Female	7 (44)
Male	9 (56)
Age, mean + range (years)	57 (36–78)
Number of years employed with the sports club, mean + range (years)	11 (1/6–25)
Number of years involved in program, mean + range (years)	8 (2–15)
Function within sports club (*n*, %) *	
Trainer	11 (69)
Board member	4 (25)
Chairperson	2 (13)
Secretary	1 (6)

* A respondent could have multiple functions within the sports club.

**Table 3 ijerph-18-06082-t003:** Descriptive characteristics of included sports clubs.

Sports Club	Sporting Program	Size Sports Club ^1^	Location of Sports Club in the Netherlands (Region)	Average Number of Participants in Sporting Program per Year (*n*)	Average Number of Participants in Sporting Program That Becomes Member of the Sports Club per Year (*n*, %)
1	Yakult Start to Run	Large	East	50	25 (50)
2	Judo in School	Small	East	1625	25 (2)
3	Through 4 Days Marches	Large	West	200	200 (100) *
4	Flexible	Medium	North	90	20 (22)
5	Flexible	Small	South	13	13 (100) *
6	Fit Hockey	Large	East	33	33 (100) *
7	Fit Hockey	Large	North	18	18 (100) *
8	Thinking and Doing	Medium	East	78	Unknown
9	Thinking and Doing	Medium	East	105	33 (31)
10	Start2Bike	Medium	North	24	20 (83)
11	Start2Bike	Medium	West	26	8 (31)
12	Cycle & Enjoy Nature	Medium	West	6	6 (100) *
13	Cycle & Enjoy Nature	Large	West	140	33 (24)
14	Trio-Triathlon	Small	South	120	Unknown
15	Cool Moves Volley	Medium	South	40; the number of participants in school clinics is unknown	40 (100) *; it is not known which percentage of participants in school clinics becomes member of the club
16	Cool Moves Volley	Large	North	45	45 (100) *

^1^ Size sports club: small: ≤100 club members; medium: 101–300 club members; large: ≥301 club members. * It is mandatory that participants become member of the sports club.

**Table 4 ijerph-18-06082-t004:** Levels of the sports club involved in providing the sporting program six and a half years after a funding period.

Level of Sports Club Involved	Sports Clubs (*n*)
Micro-level	16
Meso-level	6
Macro-level	12
Meso- and macro-level	6

**Table 5 ijerph-18-06082-t005:** Stimulating and hindering factors to involve the meso- and macro-level of the sports club in the program.

Level of Sports Club	Stimulating Factors	Hindering Factors
**Meso**	Social support for the program (SO) Enough human resources (EC) Enough financial resources (EC)	Lack of human resources (EC) Lack of financial resources (EC)
**Macro**	Feeling of social responsibility (CU) Specific focus on inactive people (CU) Multiple benefits club (SO)	No specific focus on inactive people (CU) Lack of benefits for the club as a whole (SO)

CU = cultural factor; SO = social factor; EC = economic factor.

## Data Availability

The data presented in this study are available on request from the corresponding author. The data are not publicly available due to the collection method and to preserve the privacy of respondents.

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
