# Peer review of "Can Health-Enhancing Sporting Programs in Sports Clubs Lead to a Settings-Based Approach? An Exploratory Qualitative Study"

_ijerph, 2021, doi:10.3390/ijerph18116082_

Round 1

Reviewer 1 Report

Title

  1. Do you think is the title in fit with the rest of the paper? The study used a qualitative approach? Don’t you think?

Perhaps the words as ‘exploratory’ … should be considered.

  1. Materials and Methods
  2. a) 101 102 Do you mean ‘longitudinal study…’?

‘This is a practice-based study consisting of Dutch sports clubs that have been providing sporting programs for inactive people over a period of multiple years.’

  1. b) I have missed the type of design. Please, should be included.

2.3. Data analysis

159 I strongly suggest you include some kind of statistical test.

  1. Discussion 330

4.1 From my point of view  ‘results’ should not be repeated over and over again.

4.2 I suggest you to improve the practical application of the study.

Reviewer 2 Report

The topic of the manuscript is suitable for readers of IJERPH. The paper is well written, and writing is clear. Below are my concerns/suggestions and hope these will be helpful.

- The motivation for the study should be strengthened. While previous studies generally used the top-down approach, where they looked at the process of macro-level -> meso-level -> micro-level, the authors argued that the opposite might be true and not enough research hasn’t been done (line 92). Here, the authors should provide more justifications/theories/literature on why such a bottom-up approach is possible and why looking at it bears importance for the literature and practice.

- It would be helpful if the authors provide formal definition of “sports club” in the manuscript.

- More specific and actionable practical implications should be added in the Discussion. With the findings, how do the authors help practitioners implement relevant policies?
